# Selecting optimal software code descriptors— The case of Java

**Yegor Bugayenko**[1], **Zamira Kholmatova**[2]*, **Artem Kruglov**[2], **Witold Pedrycz**[3], **Giancarlo Succi**[4]

1 Huawei, Moscow, Russia, 2 Innopolis University, Innopolis, Russia, 3 University of Alberta, Edmonton, Canada, 4 Università di Bologna, Bologna, Italy

* z.kholmatova@innopolis.university

## Abstract

Over the last 25 years, a considerable proliferation of software metrics and a plethora of tools have emerged to extract them. While this is indeed positive concerning the previous situations of limited data, it still leads to a significant problem arising both from a theoretical and a practical standpoint. From a theoretical perspective, several metrics are likely to result in collinearity, overfitting, etc. From a practical perspective, such a set of metrics is difficult to manage and companies, especially small ones, may feel overwhelmed and unable to select a viable subset of them. Still, so far it has not been fully understood what is a viable subset of metrics suitable to properly manage software projects and products. In this paper, we attempt to address this issue. We focus on the case of programs written in Java and we consider classes and methods. We use Sammon error as a measure of the similarity of metrics. Utilizing both Particle Swarm Optimization and Genetic Algorithm, we adapted a method for the identification of a viable subset of such metrics that could solve the mentioned problem. Furthermore, we experiment with our approach on 800 projects coming from GitHub and validate the results on 200 projects. With the proposed method we got optimal subsets of software engineering metrics. These subsets gave us low values of Sammon error at more than 70% at class and method levels on a validation dataset.

## 1 Introduction and motivation

The relevance of software metrics has been perceived from the very beginning of software engineering and already more than 50 years ago their benefits were considered multifaceted [1]. Initially, the problem was the lack of sound tools to collect such metrics, and the limited availability of industrially relevant metrics was considered a major impediment to the progress of the discipline [2]. Computing metrics was a complex task involving tools that were expensive and/or required advanced knowledge of programming language semantics such as the pioneering tools/configuration languages by Müller [3, 4] or Devanbu (Gen++ and Genoa) [5]. Nowadays, thanks to the intensive research and development effort in this area led by the work of international research groups, such as ISERN centered on the research of Basili [6, 7] and Rombach [8], and by the tools that have been developed in the last 25 years [9, 10]

**Data Availability Statement:** https://github.com/ZamiraKholmatova/ReducingNumberofMetrics/.

**Funding:** This research has been financially supported by The Analytical Center for the Government of the Russian Federation (Agreement

No. 70-2021-00143 01.11.2021, IGK
000000D730324P540002).

**Competing interests:** One of the authors
(Giancarlo Succi) is a Board member of PLOS ONE
journal.

(including open source tools [11]), we are in a completely different situation. It became possible to calculate the huge range of metrics such as the amount of contribution of each developer, the number of changes to the files, and many many more.

However, the possibility of considering a huge number of software metrics gave rise to the following problems:

1. Using too many metrics exposes the users to the problem of the validity of statistical analysis. For example, collinearity between metrics increases the variance among them, and, thus, leads to the wrong identification of dominant predictors in regression analysis [12]. Mathematically, we can write the regression problem (model) as $Y = XA + b$, where $Y$ is a response variable, $X$—a set of predictors (in our case, metrics), $A$—parameter estimates, and $b$—residual vector. We can estimate $A$ as follows: $A = (X^T X)^{-1} X^T Y$. If $X$ is nearly linearly dependent, then $X^T X$ is nearly singular [12]. Therefore, the estimation of $A$ will be unstable meaning that small changes in $X$ will cause huge fluctuations in $A$.

2. The huge volumes of data can hide the noise which may lead to incorrect results while working with statistical models. Without a feature selection mechanism, machine learning methods add small noise for every noisy variable to the result [13]. Having many noisy variables, the small noisy contributions are summed up and lead to high prediction errors.

3. Ineffective data management, including storage, preprocessing, and increasing computational complexity of operations over data [14].

Besides that, according to the parsimony principle, if we have two hierarchical structures (meaning that one of them can be obtained through another) and a statistically efficient estimation method, the simpler structure is asymptotically better [15–17].

To solve the problems of many metrics researchers have already suggested the following solutions: heuristic algorithms, chi-squared and Kolmogorov-Smirnov tests, Relief, and fuzzy-based methodology [18–20]. However, most of the experiments were conducted using supervised machine learning: the researchers used experts' opinions or pre-defined labels to evaluate the obtained results. This indeed introduces a potential threat to the soundness of the results, since there is an intrinsic potential [21, 22], and automatic labeling is problematic with metrics data [23].

The widely applied techniques like principal component analysis (PCA) are not adequate for our goals because they produce a linear combination of initial metrics most likely without a clear meaning assigned to them [24]. We are interested in methods that will help us to select the appropriate metrics without changing them.

Considering all of the above, our goals are therefore to:

$G_1$: Propose an approach that can identify a minimal subset of metrics that explain the "structure" of a software repository, given a set of metrics that describe it; here, the term "structure" refers to the geometrical relations between the classes or methods of a repository, represented by a set of metrics. Each class or method is described by a point in a space where each axis reflects a different metric. Therefore, if the points are close, it indicates that the classes or methods share the same properties (for example, the same number of variables, and lines of code), and vice versa. From a software engineering perspective, maintaining a consistent "structure" within a reduced set of metrics can enhance the efficiency of software analysis in terms of its quality attributes, such as fault-proneness, maintainability, recoverability, etc [25, 26]. These attributes can be found by grouping classes or methods based on the distances between them that we are aiming to save. Our goal is to reduce the dimensionality of the metrics while preserving these relations;

$G_2$: Validate this approach by applying it to a large set of open-source Java repositories;

$G_3$: On the base of $G_2$, focusing on Java code, define a minimal set of class metrics and method metrics.

To achieve our goals we analyzed 1000 open-source GitHub repositories. Our work is focused on repositories containing code written in Java: Java is mostly used by GitHub contributors in projects of different sizes and ages [27]. This allows us to consider the variety of repositories while collecting the data. We calculated 30 metrics at the class level and 28 metrics at the method level based on the code obtained from these repositories. After that, we started with the identification of a method that can help us to obtain the minimal sets of class and method metrics.

Even without labels, we still need an auxiliary objective, which helps us to understand how well the subset of metrics fits the whole set [28]. To quantify the goodness of our approximation we took Sammon error [29, 30] function, and, since the problem of calculating the error for all possible subsets became combinatorial, we employed particle swarm optimization (PSO) and genetic algorithm (GA). These two methods helped us to minimize the Sammon error without referring to an exhaustive search. Despite that GA is a well-established and very popular optimization strategy used in both academia and industry, PSO can yield almost the same results but with less computational expenses [31]. This fact leads us to the investigation of both techniques. Moreover, the usage of several techniques allows us to validate the obtained results by comparison and give the final answer. Thus, our contribution is in adapting the PSO and GA with Sammon error as a fitness function to the problem of metrics selection. Furthermore, we suggested the optimal sets of metrics at class and method levels and described the way of selecting them.

The theoretical implication of our research is presented by a feature selection method that can be combined with any of the machine learning models. The practical outcomes of our work are reflected by a minimal set of code metrics that can help software engineering practitioners to answer the question: what metrics does one have to select to preserve as much of the required information as possible? It is a key question in a situation where the number of metrics is not known in advance. This problem is present in systems requiring selecting features before the construction of the inference model [32]. As an example, we can consider the problem of code recommendation. Suppose, we have a set of developers' projects $X$ and a set of code fragments $Y$. Using metric $m$ from the developer's $x$ projects we can predict the complexity level of the developer's projects. In turn, code fragment $y$ can give us a feature from which we can infer the style of the code. We may not know in advance the metrics we will use and features about which we want to make an inference.

The paper is organized as follows. Section 1 presents the work done in the area of metrics selection. Section 3 contains a description of the methods used in this paper. The structure of experimentation, as well as the results, are included in Section 4. The discussion of the obtained results is presented in Section 5. The validation of our findings is described in Section 6. Section 7 explains the limitations of the conducted study. The discussion on whether we met our research goals is in Section 8. Section 9 summarizes the conclusion and presents the ideas for future work.

## 2 State of the art

The role of metrics in software engineering cannot be overstated: they help development teams to understand the interdependencies within the code, estimate the resources, prioritize slices for debugging, or identify data flow paths [33, 34]. However, the ability to compute vast

amounts of metrics may lead to a waste of effort, time consumption, or difficulty in focusing on important parts of data. As said, our first goal is to identify the smallest possible set of metrics that can still describe adequately a repository. This problem has already been approached in the past and sometimes, especially in machine learning, takes the name of "feature selection" [35]. The quality of selected features crucially affects the effectiveness of the constructed models [36]. We can overall divide the current approaches into two groups: a) studies requiring labeled data; b) studies not requiring labeled data.

Labeled data means that data has been tagged with descriptive information or classified by an expert (human or machine learning algorithm). Therefore, the studies of the first group require data to be "labeled". They try to remove some of the metrics from an initial set by assessing the performance of supervised machine learning algorithms before and after the removal, indeed aiming at minimizing the difference between the two [18, 37–45]. For example, Gao et al. [18] using the Chi-squared test, Gini index, Kolmogorov-Smirnov test, and Relief method reduced the set of 42 metrics to the set of 6 ones which improved the performance of the defect prediction model. The most important metrics were the number of distinct files, the number of different designers making changes, the deployment percentage of the module, and cyclomatic complexity. Shivaji et al. [42] also thoroughly studied the various types of feature selection methods and found that even 1 percent of the original number of metrics can achieve strong performance in bug identification in code changes.

The main disadvantage of the studies from the first group is that they require labeled data. Labeling is prone to subjectivity and variability and typically context-dependent, while we are aiming at an objective versatile approach.

The studies of the second group involve methods that reduce the number of metrics without access to labels. In such studies, the researchers compared the results of unsupervised algorithms like clustering or tried to extract topics and create knowledge graphs [46]. For example, Turan and Çataltepe [47] checked the changes in clusters before and after the application of PCA. Similarly to this, Ni et al. [48] ranked the features according to their influence on density-based clusters.

The limitation of the methods from the second group is that different performances of unsupervised learning algorithms applied to the initial and reduced sets of metrics do not mean that the reduction technique performs poorly. For example, the k-means algorithm can fail when clustering data with outliers or redundant metrics and then correctly cluster the reduced dataset. Therefore, the conclusions on the most important metrics can be inconsistent and unreliable. If we are talking about density-based clustering, there are also difficulties with the choice of appropriate density levels: the density level chosen for clustering the entire set of metrics may not fit the clustering of the reduced one [49]. For instance, we can find several clusters on a 3d swiss roll depending on a density level. Using the Isomap algorithm we can get the roll's 2d representation—a rectangle shape. Density-based methods find the only cluster for such a distribution of points in space. Moreover, clustering the sparse data can be even impossible: since most of the techniques rely on distances, the researchers face computational complexity while finding the distances between all the points in high-dimensional space. Other techniques described in these studies like PCA are also not in the scope of our problem since PCA-based techniques produce linear combinations of metrics as observations of a lower-dimensional space. In most cases, such linear combinations are difficult to interpret.

In turn, we are considering a very restricted case: we analyze a huge number of repositories, and each of them can be represented by a large set of metrics. Here, we should pay attention of the readers, that we are examining the scenario where metrics are computed for every repository individually at different levels (class, method, etc.). Therefore, the dimensionality reduction process will be executed independently for each repository.

Our goal is to determine how to reduce the dimensionality of vectors representing the repository, while still preserving all the meaning of the original data. There exist methods that try to nonlinearly preserve the distances or similarities between points [50]. Examples of such techniques are multidimensional scaling (MDS) [51], Isometric mapping (Isomap) [52], Sammon mapping [30]. Moreover, all the mentioned approaches can be adapted to the feature selection problem: Kruskal's stress and Sammon error—the objective functions from these algorithms are used to measure how well the selected set of features fits the whole one [53–56]. The researchers quantify the distances between samples in two distinct spaces: one formed by the complete set of features, and the other by a selected subset of features. The resulting distances are substituted into the objective function. A value near zero for this function indicates that the chosen subset of features is capable of preserving the original dataset's topology.

However, to identify the subset that produces the minimum value, an exhaustive calculation of the objective functions is necessary for all potential subsets. Calculating Kruskal's stress or Sammon error for all possible subsets of given metrics is a combinatorial problem. The most popular optimization technique—genetic algorithm showed good performance in solving combinatorial problems [57]. Also, another approach called particle Swarm optimization (PSO) was observed as an efficient technique for solving optimization problems in discrete space [58, 59]—the same task we are trying to solve. However, the findings of different researchers are ambiguous: PSO was proven to be more computationally efficient [58, 59] while GA demonstrates significantly better results in terms of convergence [57].

Both PSO and GA have been already employed in feature selection algorithms for software engineering problems, such as defect prediction [60–63], effort estimation [64], test paths identification [36], and refactoring detection [65]. However, in our work, instead of minimizing the prediction error we aim to preserve the structure of the initial data. Comparing the two selected optimization strategies, PSO is simpler to implement and faster in terms of computation time [66], while GA shows less tendency to premature convergence [67, 68]. PSO produces solutions much closer to each other thus often tending to local extremum [67, 68]. In genetic algorithms, the solutions usually cluster near several "good" points. This trade-off between computational efficiency and performance led us to study both strategies in our work.

## 3 Methodology

We represented each repository as a vector of reals:

$$X = (x_1, x_2, \cdots, x_n)^T,$$

where $x_n = (x_{n1}, x_{n2}, \ldots, x_{nk})$ is a set of metrics for a particular class or method $n_k$ of a repository $X$.

Our goal is to select such a set of metrics that will still preserve the geometrical distances between points (classes or methods). In the previous section, we observed that as an objective function of our proposed approach can be considered objective functions of existing techniques like MDS, Isomap, or Sammon mapping. MDS and Isomap have the same objective function named Kruskal's stress:

$$S = \sqrt{\frac{(d_{ij} - d_{ij}^*)^2}{d_{ij}^2}}, \tag{1}$$

where $d_{ij}$ is the distance between $i$-th and $j$-th points (classes or methods) in the higher dimensional space (represented by the whole set of metrics), $d_{ij}^*$ is the distance between the same

points in the lower dimensional space (represented by some subset of metrics). The lower the value of the function, the more the subset of metrics corresponds to the initial set.

The difference between MDS and Isomap is that MDS uses the Euclidean distance while the Isomap—geodesic.

The objective function of Sammon mapping is a weighted version of Kruskal's stress (more details are in the subsection below) [52]. Therefore, we have selected only this function to be minimized in our experiments.

However, since the calculation of Sammon error for all the possible subsets is a combinatorial problem, to find the minima of this function we need to refer to the optimization techniques. From the analysis of the literature, we discovered that PSO and GA were utilized to address this problem [69, 70]. Therefore, we also employed these techniques.

Overall the proposed methodological pipeline consists of the following parts:

1. data collection (selection of repositories and calculation of metrics);

2. finding subsets of metrics with the lowest Sammon error using PSO;

3. aggregation of the results obtained by PSO with Sammon error;

4. finding subsets of metrics with the lowest Sammon error using GA;

5. aggregation of the results obtained by GA with Sammon error.

The presented pipeline is also depicted in Fig 1.

We conducted steps 2–5 separately for metrics at class and method levels. Additionally, we performed these steps with varying sizes of metrics subsets, thus, analyzing the ability of subsets containing very few number metrics to preserve the geometrical structure.

More details on the algorithms applied in our research are given in the following subsections.

### 3.1 Data collection

We wanted to focus on the repositories that are used by many people. First of all, as a source of our data, we considered Github—the largest service for software development. As a language for our analysis, we consider Java—one of the most popular languages on GitHub [27]. Moreover, Java was the main language for the biggest projects that brought us widely used products like IntelliJ IDEA [71], and Apache Hadoop [72]. Also, to ensure that the repository is popular and still developing, we set the threshold for the number of stars and commits it should have. Overall we elaborated on the following criteria:

- code in this repository is written in Java;

- the repository contains no less than 10 commits;

- the repository has at least 100 stars;

- the project includes only open-source libraries.

Using the criteria, we have collected 1000 repositories (the list of these repositories is available through the link), which is 4.5% of the open and unarchived repositories with at least 100 stars [73]. The collected set covered the most popular GitHub topics like "java", "spring-boot", "redis", "docker", "machine-learning".

To calculate code metrics we employed SourceMeter [74]. This tool supports the calculation of more than 60 code metrics related to size, objects, and complexity. Using the taxonomy proposed by Fenton and Pfleeger [75] we were interested in internal product metrics describing

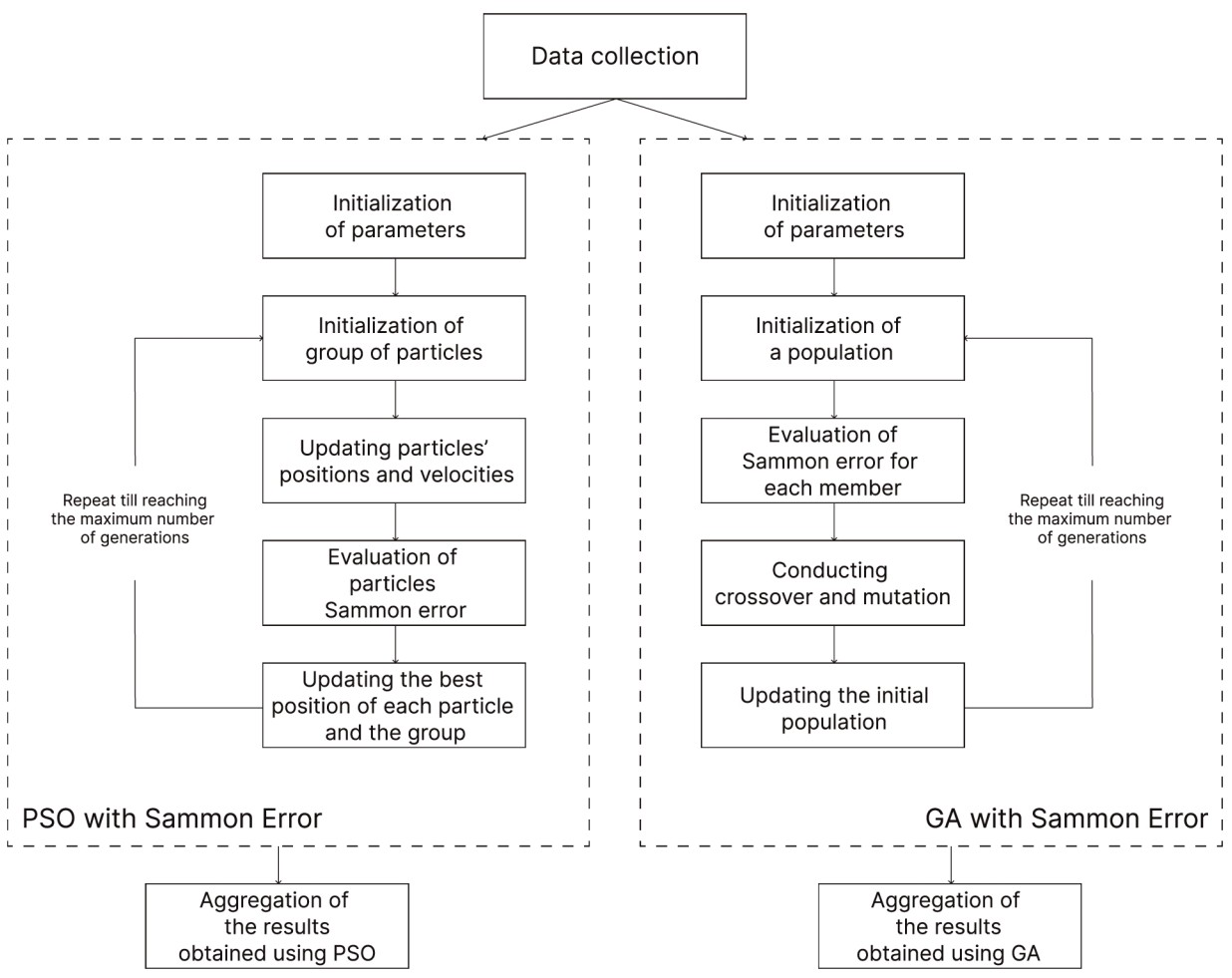

**Fig 1. Scheme of the proposed methodological pipeline.** The diagram shows the main stages of the presented methodology. The pipeline starts with the data collection process and then proceeds to the application of PSO and GA for identifying subsets of metrics that minimize Sammon error, followed by the aggregation of the obtained results.

the structure of a code at class and method levels. These metrics give us an understanding of the cohesion, complexity, coupling, documentation, inheritance, and size of a code (Tables 1 and 2).

## 3.2 Dimensionality reduction

**3.2.1 Sammon error function.**   The straightforward linear projection, such as PCA [47], tries to keep the same amount of the original variance, but not the behavior of complex patterns over data. Sammon [30] proposed an algorithm that is aimed at preserving the structure presented by $n$ points in $k$-dimensional space by finding the same number of points in $l$-dimensional space [76].

For example, $X$ is a set of input vectors from space of dimensionality $k$ and $Y = (y_1, y_2, \ldots, y_n)^T$, where $y_n = (y_{n1}, y_{n2}, \ldots, y_{nl})^T$, is a set of vectors we need to find in $l$-dimensional space.

**Table 1. Class metrics and their abbreviations.**

| Metric name | Abbreviation | Category |
|---|---|---|
| Lack of Cohesion in Methods | LCOM | Cohesion |
| Nesting Level | NL | Complexity |
| Nesting Level Else-If | NLE | Complexity |
| Weighted Methods per Class | WMC | Complexity |
| Coupling Between Object classes | CBO | Coupling |
| Coupling Between Object classes Inverse | CBOI | Coupling |
| Number of Incoming Invocations | NII | Coupling |
| Number of Outgoing Invocations | NOI | Coupling |
| Response set For Class | RFC | Coupling |
| Comment Density | CD | Documentation |
| Comment Lines of Code | CLOC | Documentation |
| Depth of Inheritance Tree | DIT | Inheritance |
| Number of Ancestors | NOA | Inheritance |
| Number of Children | NOC | Inheritance |
| Number of Descendants | NOD | Inheritance |
| Number of Parents | NOP | Inheritance |
| Lines of Code | LOC | Size |
| Logical Lines of Code | LLOC | Size |
| Number of Attributes | NA | Size |
| Number of Methods | NM | Size |
| Number of Public Attributes | NPA | Size |
| Number of Public Methods | NPM | Size |
| Number of Statements | NOS | Size |
| Total Lines of Code | TLOC | Size |
| Total Logical Lines of Code | TLLOC | Size |
| Total Number of Attributes | TNA | Size |
| Total Number of Methods | TNM | Size |
| Total Number of Public Attributes | TNPA | Size |
| Total Number of Public Methods | TNPM | Size |
| Total Number of Statements | TNOS | Size |

To find $Y$, Sammon proposed to minimize the following expression called Sammon error:

$$E = \frac{1}{\sum_{i<j} d_{ij}} \sum_{i<j} \frac{(d_{ij} - d_{ij}^*)^2}{d_{ij}}, \tag{2}$$

where $d_{ij}$ is the distance between $x_i$ and $x_j$, $d_{ij}^*$ is the distance between $y_i$ and $y_j$.

In our case, we want to approximate the interpoint distances between classes or methods by selecting a smaller set of metrics. So we select $l$, Â ($l << k$), columns (metrics) from matrix $X$ and calculate the Sammon error between $X$ and the matrix with a reduced number of metrics. Since we do not have prior knowledge of the data we have collected, we will use Euclidean distance in the Sammon function [30].

**3.2.2 Interpreting the numeric values of Sammon error function.** As we mentioned, every class or method of a repository is described by a set of $k$ metrics, thus, allowing us to represent it as a point in $k$-dimensional space. The distances between points reflect the relationships between them: if two points are close to each other, they share almost the same

**Table 2. Method metrics and their abbreviations.**

| Metric name | Abbreviation | Category |
|---|---|---|
| Lines of Duplicated Code | LDC | Size |
| Logical Lines of Duplicated Code | LLDC | Size |
| Halstead Calculated Program Length | HCPL | Complexity |
| Halstead Difficulty | HDIF | Complexity |
| Halstead Effort | HEFF | Complexity |
| Halstead Number of Delivered Bugs | HNDB | Complexity |
| Halstead Program Length | HPL | Complexity |
| Halstead Program Vocabulary | HPV | Complexity |
| Halstead Time Required to Program | HTRP | Complexity |
| Halstead Volume | HVOL | Complexity |
| Maintainability Index | MI | Complexity |
| McCabe's Cyclomatic Complexity | McCC | Complexity |
| Nesting Level | NL | Complexity |
| Nesting Level Else-If | NLE | Complexity |
| Number of Incoming Invocations | NII | Coupling |
| Number of Outgoing Invocations | NOI | Coupling |
| Comment Density | CD | Documentation |
| Comment Lines of Code | CLOC | Documentation |
| Documentation Lines of Code | DLOC | Documentation |
| Total Comment Density | TCD | Documentation |
| Total Comment Lines of Code | TCLOC | Documentation |
| Logical Lines of Code | LLOC | Size |
| Lines of Code | LOC | Size |
| Number of Statements | NOS | Size |
| Number of Parameters | NUMPAR | Size |
| Total Logical Lines of Code | TLLOC | Size |
| Total Lines of Code | TLOC | Size |
| Total Number of Statements | TNOS | Size |

properties (e.g. two classes have the same number of methods), and vice versa. We are aiming at reducing the number of metrics by preserving the relationship between methods or classes. Sammon error quantifies such preservation. If we consider the classes $c_1 = (4, 6, 3)$ and $c_2 = (4, 18, 9)$ represented by three metrics, the removal of the first metric will not significantly affect the Euclidean distance between $c_1$ and $c_2$ since it is the same for both classes. Therefore, to minimize Sammon error we will keep metrics 2 and 3 meaning that various metrics (having different values for different classes) provide us with more information while describing a software repository.

## 3.3 Optimization strategies

**3.3.1 Particle swarm optimization.** PSO is one of the optimization algorithms used in the proposed approach [69]. In this algorithm, the particle represents the set of features. Each feature is encoded with a number. For example, we use the floating-point encoding to control the number of features in a subset. During each iteration, each particle moves to its previous best position $pbest_i$ and to the global best position $gbest$. This movement occurs due to the updating

of the velocity $v_i$ and position $e_i$ of each particle [69]:

$$v_i = wv_i + c_1r_1(pbest_i - e_i)+$$
$$c_2r_2(pbest_i[gbest] - e_i),$$
$$e_i = (e_i + v_i),$$

(3)

where $i$ is the particle's number, $w$—an inertia weight, $r_1$ and $r_2$—two independent random vectors in (0, 1), $c_1$ and $c_2$—positive constants called cognitive and social learning features.

**3.3.2 Genetic algorithm.** The second algorithm that can be used to minimize the Sammon error is GA. GA creates the initial population and then tries to improve it through evolution [70]. The evolution is usually done via parent selection, crossover, and mutation. Each member of a population represents a set of features. Similar to PSO, one should consider floating-point encoding of features in each member of a population to control the number of features included in the finite subset [77]. Float encoding allows ranking features and then selecting the required number of them to calculate the fitness function. In each iteration, we have conducted a one-point crossover of parents with probability $p_c$ to obtain two children and a mutation with a probability $p$ on every child.

## 3.4 Aggregation of the results

To aggregate the results, we applied a technique similar to vote-counting in meta-analysis [78, 79]. Vote-counting implies the comparison of the number of studies with significant positive results and significant negative results; in simple words, if most of the studies that examined the same effect yielded positive significant results, then this effect is considered to have a positive effect, and vice versa. In our case, we identified optimal subsets of different sizes as those comprising the most frequently occurring metrics.

## 4 Results of experiments

For the selected with our criteria 1000 projects we have calculated 30 metrics at a class and 28 ones at a method levels. Then we divided the collected data into train and validation sets in the proportion of 80/20. The descriptive statistics of both sets are provided in Table 3.

We used a train set to conduct all the steps of our methodological pipeline to find optimal subsets of metrics at class and method levels. To understand whether the obtained subsets give the same results we employed the validation set.

The values of the hyperparameters of PSO and GA used in the experiments are summarized in Table 4.

Firstly, we applied PSO to minimize Sammon error (Eq (2)) separately for metrics at class and method levels using the train set. We conducted experiments on each repository, varying the number of metrics at both class (ranging from 2 to 29) and method (ranging from 2 to 27) levels. Each experiment was repeated 3 times. The final result in every repository for each number of metrics was determined by selecting the smallest recorded value. The minimum, average, and maximum values of Sammon errors for each number of metrics across the train set of repositories at both levels are presented in Figs 2 and 3. Tables A1 and A3 in

**Table 3. Train and validation set description.**

| Dataset | Total # of repositories | Total # of classes | Total # of methods |
|---|---|---|---|
| Train | 800 | 155240 | 720107 |
| Validation | 200 | 43045 | 205948 |

**Table 4. PSO and GA hyperparameters.**

| Parameter name | PSO | GA |
|---|---:|---:|
| Population size | 20 | 20 |
| Number of iterations | 20 | 20 |
| Inertia weight | 0.6 | - |
| First cognitive vector | 2 | - |
| Second cognitive vector | 2 | - |
| First social vector | 2 | - |
| Second social vector | 2 | - |
| Probability of a crossover | - | 0.6 |
| Probability of a mutation | - | 0.6 |

S1 Appendix show the number of occurrences of each metric in subsets of 2 to 20 metrics at the class and method levels, respectively.

We also applied GA to find the minimum value of Sammon error (Eq (2)). During each iteration, we randomly selected pairs to produce a new generation. Each pair passed a single-point crossover and their children underwent a mutation. Then the population was replaced with children. Similar to the PSO approach, we conducted experiments for each repository across different numbers of metrics at both class and method levels, repeating the process three times. The outcome was determined by selecting the smallest Sammon error value obtained. The results are presented in Fig 4, Table A2 in S1 Appendix (the number of appearance of each metric in subsets of 2 to 20 class metrics), Fig 5, and Table A4 in S1 Appendix (the number of appearance of each metric in subsets of 2 to 20 method metrics).

The statistics of time spent to find the optimal subset in every experiment with the parameters above are provided in Table 5.

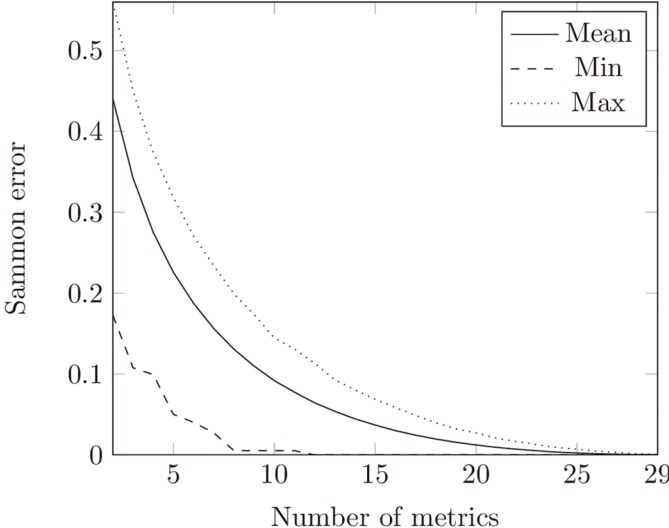

**Fig 2. Sammon error obtained using PSO for metrics at the class level.** The figure shows the minimal, mean, and maximum values of Sammon error obtained from the PSO optimization runs. The X-axis indicates the number of metrics in the potential optimal subsets at the class level, while the Y-axis represents the corresponding values of Sammon error.

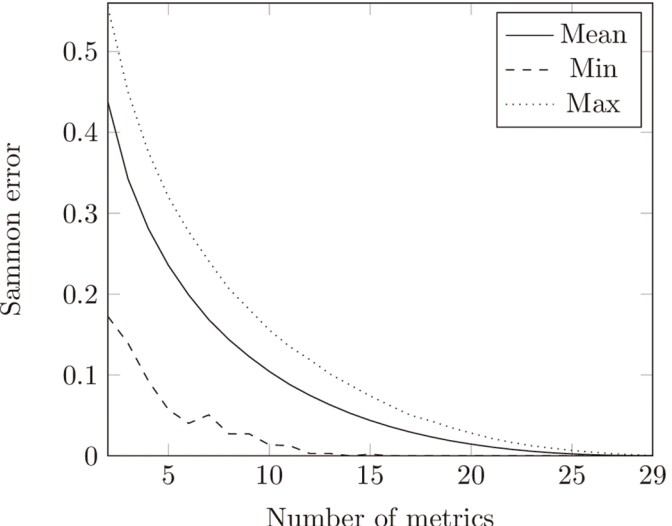

**Fig 3. Sammon error obtained using PSO for metrics at the method level.** The figure shows the minimal, mean, and maximum values of Sammon error obtained from the PSO optimization runs. The X-axis indicates the number of metrics in the potential optimal subsets at the method level, while the Y-axis represents the corresponding values of Sammon error.

All the experiments were conducted on a hardware with the central processing unit AMD RYZEN Socket AM4 X8 R7–5800X with 32 Gb of RAM and graphical processing unit TUF-RTX3080TI-12G-GAMING with 12 Gb of VRAM.

## 5 Discussion

Comparing Fig 2 with Fig 4 and Fig 3 with Fig 5, one can notice that PSO and GA give almost the same values of Sammon error. However, we compared distributions of values obtained

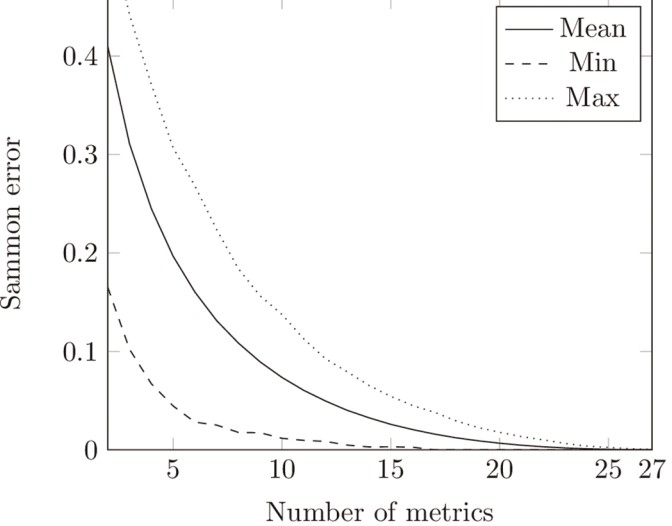

**Fig 4. Sammon error obtained using GA for metrics at the class level.** The figure shows the minimal, mean, and maximum values of Sammon error obtained from the GA optimization runs. The X-axis indicates the number of metrics in the potential optimal subsets at the class level, while the Y-axis represents the corresponding values of Sammon error.

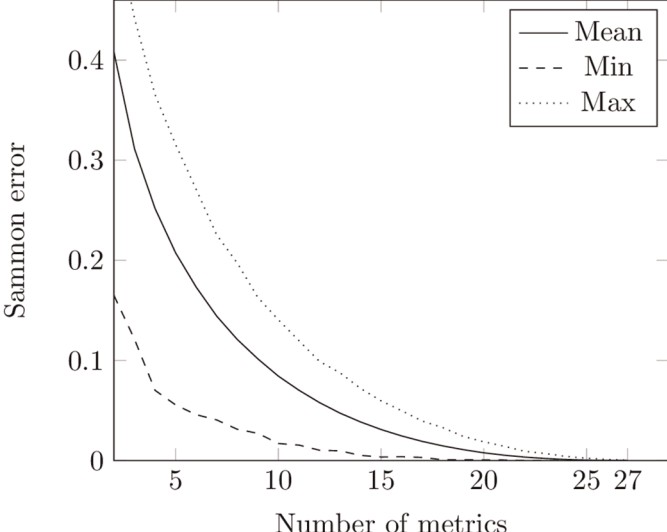

**Fig 5. Sammon error obtained using GA for metrics at the method level.** The figure shows the minimal, mean, and maximum values of Sammon error obtained from the GA optimization runs. The X-axis indicates the number of metrics in the potential optimal subsets at the method level, while the Y-axis represents the corresponding values of Sammon error.

through these two techniques for the same number of features using the Mann-Whitney test (the non-parametric counterpart of the independent t-test) [80, 81]. The null hypothesis posits that the distributions of Sammon errors derived from PSO and GA are identical, whereas the alternative hypothesis asserts that the distribution of values obtained through PSO is lower than those obtained through GA. At a class level, the was no difference in usage of PSO and GA when the number of metrics was equal to 2, 3, 26, 27, 27. For other numbers of metrics, the test showed that the distribution underlying errors obtained by employing PSO is less than the one obtained by employing GA. No difference in usage PSO and GA was also observed for optimal sets of 2, 3, 24, 25, and 26 metrics at a method level. In other cases, the test showed the same trend as in a class level.

Nevertheless, remains the question about the identification of Sammon error values that characterize a chosen subset of metrics as "good". Sammon errors below 25% have been deemed low in visualization tasks [82]. Having no other information at hand, we also defined the subset as optimal when it yielded an error value of less than 0.25.

In our experiments with the Sammon error function on the train set, both PSO and GA gave values less than 0.25 starting from 7 metrics at the class and 6 metrics at the method levels (see S1 Appendix).

The subset of 7 metrics (CD, NOP, NLE, DIT, CBO, NL, NA) demonstrates the lower Sammon error: 537 repositories from a train set showed a value less than 0.25.

**Table 5. Time spent for experimentation.**

| Approach | Level | Mean (s) | Min (s) | Max (s) |
|---|---|---|---|---|
| PSO | Class | 0.04040 | 0.00396 | 2.04671 |
| GA | Class | 0.04193 | 0.00495 | 2.04767 |
| PSO | Method | 0.21032 | 0.00402 | 4.88547 |
| GA | Method | 0.21171 | 0.00505 | 4.85664 |

**Table 6. Mean and standard deviation of Sammon error obtained for the optimal sets of metrics.**

| Level | Median | Mean | Std |
|---|---|---|---|
| Class (7 metrics) | 0.22 | 0.22 | 0.07 |
| Method (6 metrics) | 0.22 | 0.22 | 0.09 |

**Table 7. Execution time of PSO and GA using Sammon error at a class level.**

| Statistic | PSO (s) | GA (s) |
|---|---|---|
| Mean | 0.0403 | 0.0418 |
| Median | 0.0069 | 0.0085 |
| Standard deviation | 0.1108 | 0.1102 |
| Minimum | 0.004 | 0.0049 |
| Maximum | 2.044 | 2.0869 |

**Table 8. Execution time of PSO and GA using Sammon error at a method level.**

| Statistic | PSO (s) | GA (s) |
|---|---|---|
| Mean | 0.2102 | 0.2116 |
| Median | 0.0181 | 0.0196 |
| Standard deviation | 0.4787 | 0.4779 |
| Minimum | 0.004 | 0.0051 |
| Maximum | 4.881 | 4.8566 |

At the method level, in Tables A3 and A4 in S1 Appendix subsets of 6 metrics (NUMPAR, CD, TCD, NLE, NOI, NL) give us the Sammon error less than 0.25 on average (513 repositories from a train set showed such values).

The median, mean, and standard deviation of Sammon error calculated for the optimal set of metrics at a class level and method levels in all train repositories are presented in Table 6.

However, looking at Figs 2–5, it is also noticeable that the bottom lines in all these figures are not smooth: this can be explained by convergence to local minima of both methods. We cannot guarantee that all the values obtained in our experiments are the optimal ones.

Nevertheless, we have established that PSO provides better performance and spends a bit less time at class (Table 7) and method (Table 8) levels in comparison with GA.

## 6 Validity

To ensure the internal validity of our study, we employed two optimization strategies—PSO and GA, and compared the results obtained from both methods. The fact that the two methods produced similar results in terms of optimal subsets of metrics indicates the high degree of convergent validity in our study.

To verify whether the found subsets of metrics would still give us a Sammon error of less than 0.25, we measured the Sammon error for these subsets on the validation set of 200 repositories. For 148 repositories at a class and 146 at a method levels we got the Sammon errors less than 0.25. The median, mean, and standard deviation of the results we obtained are presented in Table 9. The results are independent of each other, implying that the calculation of error for one repository does not impact the calculations for others.

**Table 9. Mean and standard deviation of Sammon error obtained during validation.**

| Level | Median | Mean | Std |
|---|---|---|---|
| Class (7 metrics) | 0.20 | 0.21 | 0.07 |
| Method (6 metrics) | 0.20 | 0.21 | 0.09 |

Therefore, to verify that the Sammon errors computed for the 200 validation repositories were lower than those computed for the 800 training repositories, we conducted two additional hypothesis tests. At the class level, we tested the null hypothesis that the distribution of errors for the optimal subset during the train and validation are identical, again the one-sided alternative that the validation set gives lower values. Using the Mann-Whitney test, we rejected the null hypothesis with a p-value of 0.0241. Similarly, at the method level, we tested the same null hypothesis against the same one-sided alternative and also rejected the null hypothesis in favor of the alternative with a p-value of 0.0027.

The proposed approach performed well as the optimal set of metrics at both levels had a Sammon error of less than 0.25. However, the sets contained potentially correlated metrics such as NL and NLE, CD, and TCD. It is important to note that Sammon error does not account for collinearity between vectors. Overall, the validation results support both the internal and external validity of our findings.

The external validity of our research is also supported by the implication of randomization when selecting the data sample. The projects present in both the training and validation sets vary in terms of their size, allowing us to claim high representativeness of the selected sample (Figs 6–9).

## 7 Limitations

Even after considering the different types of validity our study still has some limitations that have to be acknowledged:

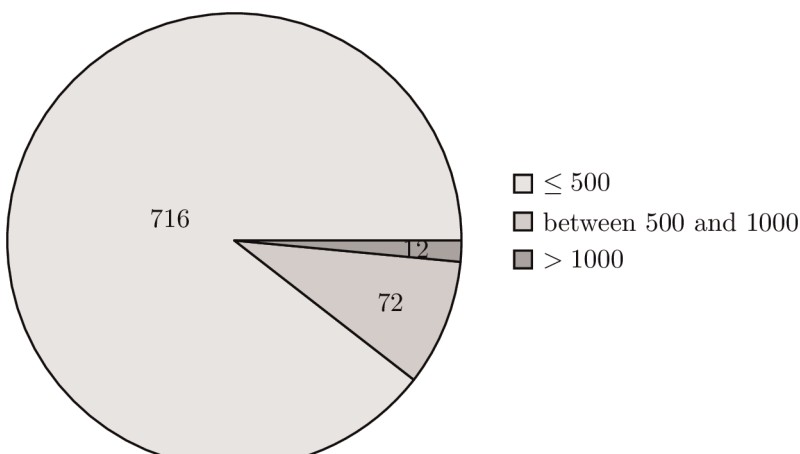

**Fig 6. Distribution of repositories in the train set by the number of classes.** The pie chart shows the distribution of repositories in the training set, categorized by the number of classes: repositories with 500 or fewer classes, those with between 501 and 1000 classes, and repositories with more than 1000 classes.

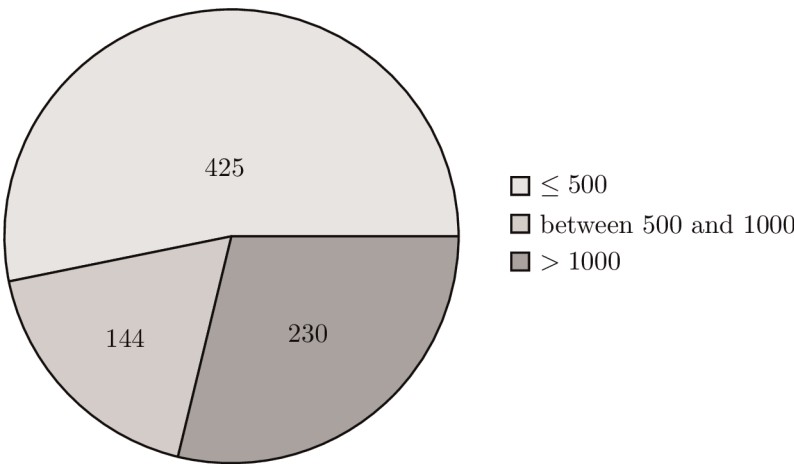

**Fig 7. Distribution of repositories in the train set by the number of methods.** The pie chart shows the distribution of repositories in the training set, categorized by the number of methods: repositories with 500 or fewer methods, those with between 501 and 1000 methods, and repositories with more than 1000 methods.

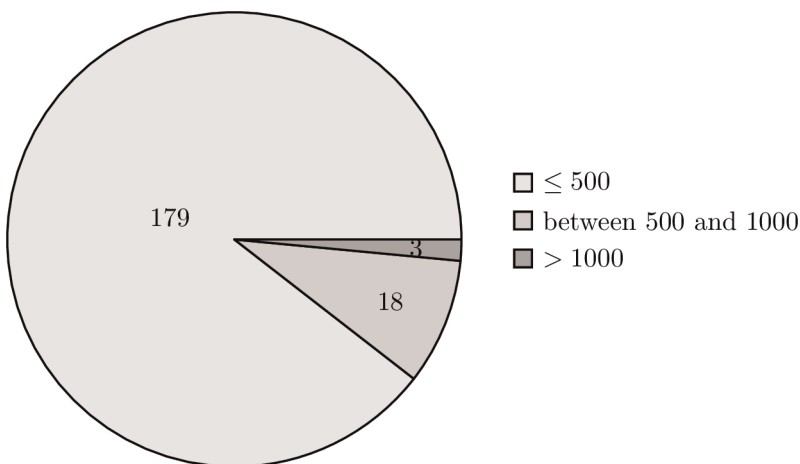

**Fig 8. Distribution of repositories in the validation set by the number of classes.** The pie chart shows the distribution of repositories in the validation set, categorized by the number of classes: repositories with 500 or fewer classes, those with between 501 and 1000 classes, and repositories with more than 1000 classes.

- choice of the language. Our study is restricted to the case of Java meaning that we need additional research to generalize our findings;

- potential collinearity of metrics. Although we have identified the optimal sets of software engineering metrics, some of these metrics seem to be collinear;

- sampling technique. Even though we tried to ensure the sample variability in terms of the size, domain, and popularity, we still cannot guarantee the extent to which the obtained sample captures the real-world scenario;

- threshold of Sammon error. The considered threshold was appropriate for our study without an implication of being suitable to other datasets;

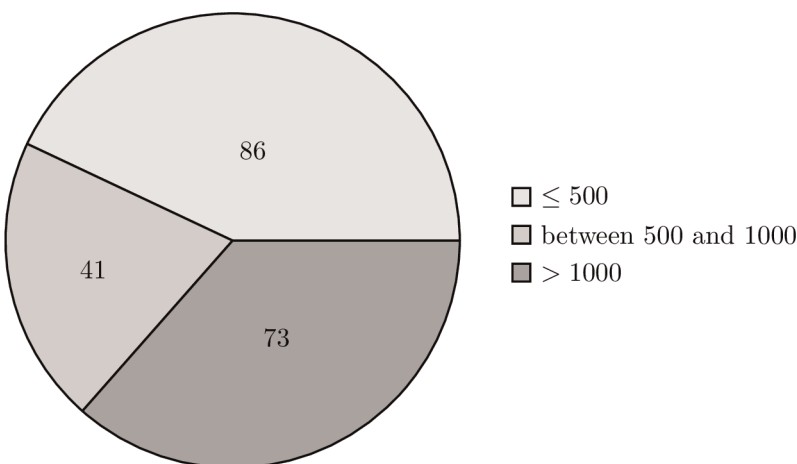

**Fig 9. Distribution of repositories in the validation set by the number of methods.** The pie chart shows the distribution of repositories in the validation set, categorized by the number of methods: repositories with 500 or fewer methods, those with between 501 and 1000 methods, and repositories with more than 1000 methods.

- choice of metrics. Our work analyzes only a limited set of metrics. There exist other metrics that could be relevant to our analysis and not included in the scope of this study.

## 8 Review of the goals of the research

### 8.1 Review of the $G_1$

The first goal of the research dealt with identifying a minimal set of metrics describing the "structure" of a repository.

Analyzing the results from the conducted experiments, we propose the following approach that for a given set of metrics identifies the minimal subset of them—PSO with Sammon error. The main idea of this approach is to preserve the Euclidean distances between classes or methods presented by metrics. Sammon error quantifies such preservation. Minimizing Sammon error using optimization strategies helps us to avoid consideration of all possible subsets. During our experiments, we have found that PSO shows good performance while minimizing Sammon error and consumes less time than GA.

### 8.2 Review of $G_2$

The second goal of the research aimed at validating the approach defined in $G_1$ on a large set of open-source Java repositories. During answering $G_1$, we run the approach for 800 open-source Java repositories at a class and method levels. The initial number of class metrics is 30, and the method metrics—28.

### 8.3 Review of $G_3$

The third goal of the research was to find, based on the result of $G_2$ and focusing on Java code, a minimal set of class metrics and method metrics properly describing *in general* a Java repository. The main obstacle we have faced during this research is the aggregation of the results obtained from each repository: how we can combine about 30 subsets from 800 projects and give a unified answer. Software engineering lacks the methods for aggregation or

generalization of the results from multiple experiments. There already exist different methods like meta-analysis or vote-counting [79, 83]. Meta-analysis is mostly used to combine correlation coefficients or mean differences which is not the scope of our work. Vote-counting is the least powerful aggregation technique that should be avoided whenever possible. Nevertheless, we tried to aggregate our results by repeating the experiments (3 times for each number of metrics in every repository) and comparing the results obtained from PSO with Sammon Error with those obtained with the help of GA. We define an optimal set of metrics for each metric count as a subset comprising the most frequently occurring metrics. This approach helped us to observe patterns in the appearance of metrics in different subsets (Tables A1–A4 in S1 Appendix) and give a common answer. Overall, as optimal sets of metrics, we can consider the following ones:

- at the class level: Comment Density, Number of Parents, Nesting Level Else-If, Depth of Inheritance Tree, Coupling Between Object classes, Nesting Level, Number of Attributes;

- at the method level: Number of Parameters, Comment Density, Total Comment Density, Nesting Level Else-If, Number of Outgoing Invocations, Nesting Level.

These subsets are obtained by analyzing classes and methods in every repository as the points in high dimensional spaces and considering the Euclidean distances between them. Exclusion of an expert's opinion or focus on the "name" of the metric prevents us from bias while running our experiments.

## 9 Conclusion

In this research, we were aimed to adapt a method for finding optimal subset of software engineering metrics. For this purpose, we have employed Sammon error as a fitness function with PSO and GA as optimizers. To run the experiments, we collected a dataset with 30 class and 28 method metrics for 1000 open-source Java repositories.

In our experiments, both PSO and GA gave almost the same results in terms of Sammon error. It could be due to the medium population size and the number of iterations in GA. This observation prompted us to compare the execution time of these methods. The results show us that PSO can be a bit faster than GA (Tables 7 and 8). Considering both the performance and the execution time, we suggest using PSO with Sammon error to select the appropriate metrics. Found by this method the optimal subsets gave us low values of Sammon error at 74% of validation repositories at a class and 73% at a method level. Moreover, the obtained subsets have intersected metrics with the subsets found in the other works. For example, Zivkovic et al. [84] suggested a software defect prediction approach based on a reptile search algorithm for optimization. The analysis of their results confirmed that nesting level, depth of inheritance tree, and the number of dependencies that the observed class owns—the metrics that also appeared in our optimal subsets, are significant for the defect prediction.

As previously noted, the Sammon error function does not account for collinearity among metrics. Consequently, we aim to address this issue in our future research by mitigating collinearity while selecting the most suitable set of metrics. Despite these limitations, our study provides valuable insights into the selection of metrics when the number of features is unknown a priori. Moreover, our findings demonstrated the potential of PSO with Sammon error to define the smaller subsets of metrics faster than GA with Sammon error.

**As a future work**, we intend to explore other **optimization strategies and fitness functions** and search for optimal parameters for these methods. We also plan to explore **novel techniques to preserve maximal information** from the complete set of metrics while maintaining the interpretability of selected features. Additionally, we aim to expand our study to

include **more programming languages and different software engineering tasks** to generalize our findings. We assume that other languages similar to Java (for example, C#) will work similarly, but at the moment we have no supporting evidence. Therefore to understand the effect of the choice of programming languages on the results we want to proceed with experiments on other languages—similar to Java, different from it (like Rust), or the most popular in Github (like Python). By considering these aspects, we believe that we can enhance the methodology for selecting software engineering metrics, ultimately improving the quality of software development.

## Supporting information

**S1 Appendix.**
(PDF)

## Author Contributions

**Conceptualization:** Yegor Bugayenko, Giancarlo Succi.

**Data curation:** Zamira Kholmatova.

**Formal analysis:** Yegor Bugayenko.

**Funding acquisition:** Giancarlo Succi.

**Investigation:** Zamira Kholmatova.

**Methodology:** Yegor Bugayenko, Zamira Kholmatova, Witold Pedrycz, Giancarlo Succi.

**Project administration:** Giancarlo Succi.

**Software:** Yegor Bugayenko, Zamira Kholmatova, Artem Kruglov, Giancarlo Succi.

**Supervision:** Artem Kruglov, Witold Pedrycz, Giancarlo Succi.

**Writing – original draft:** Zamira Kholmatova, Artem Kruglov.

**Writing – review & editing:** Witold Pedrycz, Giancarlo Succi.

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
