## [Decision Letter · Decision Letter 0]

25 Jun 2024

PONE-D-24-18681Selecting Optimal Software Code Descriptors --- the Case of JavaPLOS ONE

Dear Dr. Kholmatova,

Thank you for submitting your manuscript to PLOS ONE. After careful consideration, we feel that it has merit but does not fully meet PLOS ONE’s publication criteria as it currently stands. Therefore, we invite you to submit a revised version of the manuscript that addresses the points raised during the review process.

Comments from PLOS Editorial Office: We note that one or more reviewers has recommended that you cite specific previously published works. As always, we recommend that you please review and evaluate the requested works to determine whether they are relevant and should be cited. It is not a requirement to cite these works. We appreciate your attention to this request.

We look forward to receiving your revised manuscript.

Kind regards,

Vijayalakshmi Kakulapati, Ph.D

Academic Editor

PLOS ONE

Reviewers' comments:

Reviewer's Responses to Questions

**Comments to the Author**

1. Is the manuscript technically sound, and do the data support the conclusions?

Reviewer #1: Yes

Reviewer #2: Partly

2. Has the statistical analysis been performed appropriately and rigorously? 

Reviewer #1: Yes

Reviewer #2: Yes

3. Have the authors made all data underlying the findings in their manuscript fully available?

Reviewer #1: Yes

Reviewer #2: Yes

4. Is the manuscript presented in an intelligible fashion and written in standard English?

Reviewer #1: Yes

Reviewer #2: Yes

5. Review Comments to the Author

Reviewer #1: In the manuscript, the authors address a contemporary and interesting topic.

They applied particle swarm optimization (PSO) and a genetic algorithm (GA). However, good results have also been obtained with the techniques described in https://doi.org/10.1016/j.asoc.2023.110659. If the authors offer a solution that does not minimize the prediction error, it would be interesting for readers to see a comparison of the proposed solution with those mentioned in the paper.

How were the 1000 repositories to be analyzed determined? How was this value chosen, and does it affect the results obtained?

Sections 3.3.1 and 3.3.2 contain details about algorithms that are likely familiar to readers, so it is recommended to shorten them.

Are the results affected by the choice of programming language? How do the characteristics of the programming language impact the research? In the conclusion, the authors mention the application of their work with other programming languages, but a more detailed analysis would be useful.

Reviewer #2: 1. Describe dataset features in more details and its total size and size of (train/test) as a table.

2. Pseudocode / Flowchart and algorithm steps of the proposed method need to be inserted.

3. Time spent need to be measured in the experimental results.

4. Limitation and Discussion Sections need to be inserted.

5. The cost associated with deploying these deep learning models, including the necessary hardware and software, is not addressed.

6. The parameters used for the analysis must be provided in table

7. The architecture of the proposed model must be provided

8. Address the accuracy/improvement percentages in the abstract and in the conclusion sections, as well as the significance of these results.

9. The authors need to make a clear proofread to avoid grammatical mistakes and typo errors.

10. Add future work in last section (conclusion) (if any)

11. The authors need to add recent articles in related work and update them.

12. To improve the Related Work and Introduction sections authors are recommended to review this highly related research work paper:

a) An Approach to Slicing Object-Oriented Programs

b) An ASP .NET Web Applications Data Flow Testing Approach

c) Automatic PSO Based Path Generation Technique for Data Flow Coverage

d) A new feature selection method based on frequent and associated itemsets for text classification

e) Topic Extraction and Interactive Knowledge Graphs for Learning Resources

f) Developing an efficient method for automatic threshold detection based on hybrid feature selection approach

6. PLOS authors have the option to publish the peer review history of their article (what does this mean?). If published, this will include your full peer review and any attached files.

Reviewer #1: No

Reviewer #2: No

---

## [Author Response · Author response to Decision Letter 0]

11 Aug 2024

We would like to express our most sincere gratitude to the reviewers for the very valuable observations that they provided us, which allowed us to identify weak areas in our work and, hopefully, to improve it.

In the attached file "response_letter" there are our specific responses to them.

---

## [Decision Letter · Decision Letter 1]

8 Sep 2024

Selecting Optimal Software Code Descriptors --- the Case of Java

PONE-D-24-18681R1

Dear Dr. Kholmatova,

We’re pleased to inform you that your manuscript has been judged scientifically suitable for publication and will be formally accepted for publication once it meets all outstanding technical requirements.

Kind regards,

Vijayalakshmi Kakulapati, Ph.D

Academic Editor

PLOS ONE

**Comments to the Author**

Reviewer #1: All comments have been addressed

Reviewer #2: All comments have been addressed

Reviewer #1: All of the requirements were met by the authors. The new version of the manuscript meets the requirements for publication in this type of journal.

Reviewer #2: The updated manuscript, which addresses previous comments and suggestions, has been evaluated positively. The revised submission demonstrates significant improvement and provides valuable insights relevant to the research community. I recommend accepting it for publication.

---

## [Editor Report · Acceptance letter]

20 Oct 2024

PONE-D-24-18681R1 

PLOS ONE

Dear Dr. Kholmatova, 

I'm pleased to inform you that your manuscript has been deemed suitable for publication in PLOS ONE. Congratulations! Your manuscript is now being handed over to our production team.

Kind regards, 

on behalf of

Dr. Vijayalakshmi Kakulapati 

Academic Editor

PLOS ONE